# Wind–Wave Conditions and Change in Coastal Landforms at the Beach–Dune Barrier of Cesine Lagoon (South Italy)

**Marco Delle Rose *** and **Paolo Martano**

Institute of Atmospheric Sciences and Climate, National Research Council of Italy, 73100 Lecce, Italy; p.martano@isac.cnr.it
* Correspondence: m.dellerose@le.isac.cnr.it

**Abstract:** Several coastal barriers experienced significant erosion and change in shape throughout the Mediterranean coasts over the past decades, and the issue has become of increasing concern for scientists and policymakers. With reference to a case study and by meteorological and geomorphological investigations, this note aims to define the wind–wave conditions, infer the net longshore transport, and detect the geomorphological processes that shape the landforms of the Cesine Lagoon barrier (South Italy). Despite the importance of the site in coastal defense and environmental conservation, there are still no specific studies. A challenge for this research was to obtain significant results from publicly available sources and simple and inexpensive methods. Geomorphological changes, such as the retreat of dune toes, accretion of washover fans, and formation of gravel beaches, are related to the analyzed wind–wave conditions. The net longshore transport is found in accordance with the direction of the more intense winds. The role of extreme events in the shaping of coastal landforms is yet to be established, even if they greatly increase the vulnerability to flooding of the study area. The results achieved so far are starting points for further data collection and analysis in the perspective of assessing the impact of climate changes and the threatening hazards on the lagoon barrier.

**Keywords:** wind statistics; wave height; coastal storm; multi-temporal image analysis; geographic object; longshore transport; shoreline dynamics; washover fan

## 1. Introduction

Lagoon barriers are highly vulnerable landform systems that change and move landwards or seawards at different timescales as a result of coastal hydrodynamics, sea level change, sediment supply, geological setting, and human intervention. They are ecologically essential because they act as buffer zones for the nearby wetlands, safeguarding their ecosystems from coastal winds, waves, and storm surges as well as serving as habitats for resident and migratory bird populations. Moreover, as for other types of coastal barriers, their conservation and stabilization are essential for the protection of buildings and roads in many locations of the world [1–4].

The degrees of change in the coastal geomorphology and shoreline position are crucial in hazard zoning, hydrodynamics analysis, sediment budgeting, morphodynamics modeling, and coastal settlement policy [5,6]. Some coastal landforms, such as beaches, dunes, and washover fans, are particularly sensitive to the wind–wave conditions. Consequently, short-term studies can allow to focus on the weather factors that control their change, thus establishing the basis for subsequent medium–long-term analyses [7–9].

Along the Mediterranean coasts, several barriers experienced significant erosion and change in shape over the past decades, and the issue has become of increasing concern for scientists and policymakers [10–13]. With reference to a case study, the beach–dune barrier of Cesine Lagoon (Apulia region, South Italy), this note aims to (1) define the wind–wave conditions, in which geomorphological changes occur; (2) infer the direction of the net longshore transport of sediment; and (3) detect the geomorphological processes

that shape coastal landforms, suggesting relations with wind and wave parameters. The study was performed between 2015 and 2021 because of the available field observations and high-quality satellite images. The choice of the case study is motivated by its strong geomorphological change and shoreline recession, both of which have occurred at least since the 1980s [12,14]. A challenge for this research was to obtain significant results from publicly available sources and simple and inexpensive methods. Actually, the achieved findings may be considered starting points for further data collection and analysis.

## 2. Study Area

### 2.1. Geological–Environmental Setting

The study area is located in the Southeastern Apulia region on the western side of the Adriatic Sea, the northernmost arm of the Mediterranean Sea (Figure 1). It has the following oceanographic characteristics: mean tidal range (the difference between the average high tide level and the average low tide level) of 0.3 m, annual closure depth (the theoretical depth where sediment transport is negligible in a one year interval) around 5.5 m, and storm–wave base level (the depth beyond which storm wave action ceases to stir the sediments) between 20 and 25 m [15–17]. The Cesine Lagoon is an ensemble of water bodies, the greatest and more ecologically important of which is named "Pantano Grande" (that literally means "large quagmire" in Italian). It has a different protection status, such as WWF Nature Reserve, Natura 2000 Area, and RAMSAR Site (no. 168), and is also a regional geosite [18].

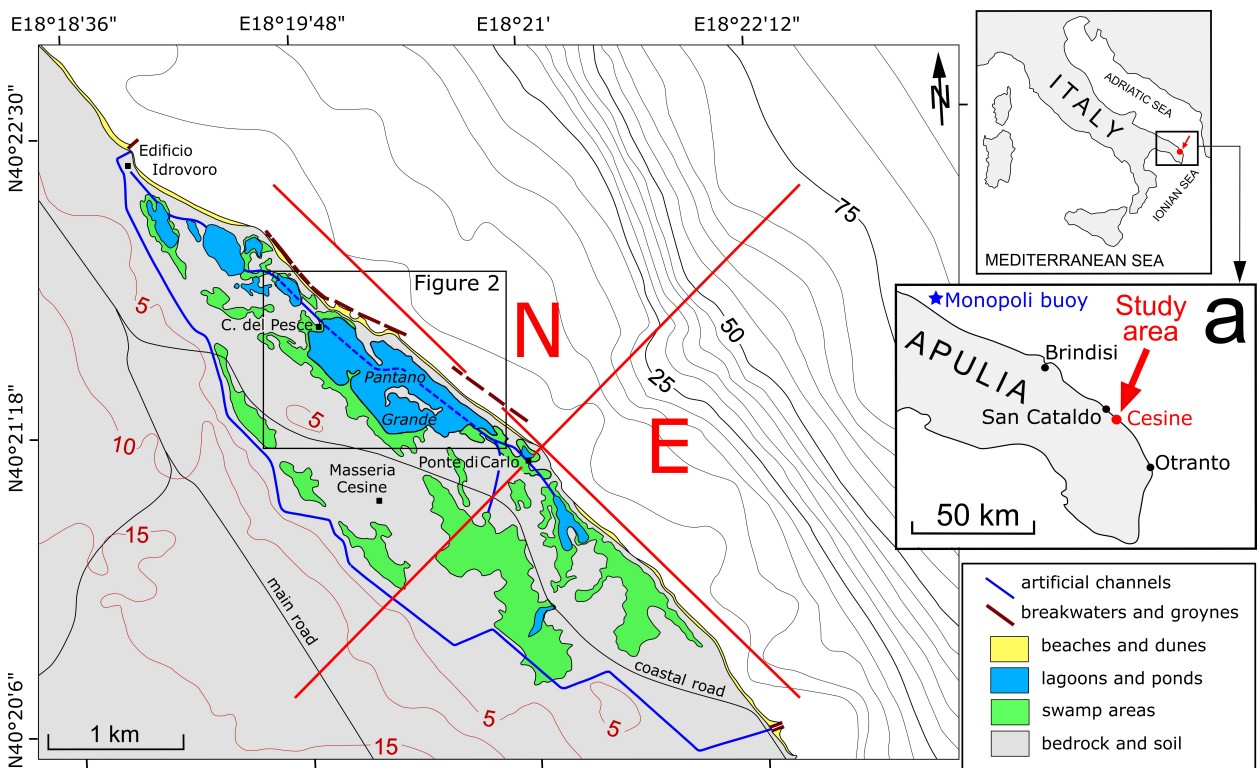

**Figure 1.** Map of the study area (contour lines and isobaths are reported; contour interval 5 m). Wind direction sectors (see text) are subdivided by crossed red lines (N = North, E = East). (**a**) Localization of Monopoli buoy and coastal towns where wind gauge stations are placed (see text).

The water lagoon depth is 0.8 m on average, shoreward reduced up to 0.3 m by silting processes [19]. Several swamps border the water bodies, forming, as a whole, a large wetland. Before environmental protection measures were in place, the dune ridges were exploited for sand mining throughout the southern Apulia coast. The mined sand was used for ship ballasting from the XIX century; for swamp reclamation in the first half of

the XX century; and for concrete production up to the 1970s. Such human activities likely determined a loss of sand comparable to that removed by natural processes, while the dune morphology was considerably altered [12]. As a matter of fact, the dune vegetation of the Cesine Lagoon barrier system is currently affected by the loss and fragmentation of habitat types, and characterized by a pioneer vegetation including sand couch grass (*Elytrigia juncea*) [20,21].

The stretch of gently sloping coast facing the Cesine Lagoon belongs to a physiographic unit that extend for 70 km from Brindisi to Otranto (see panel a of Figure 1) [15,22]. It is oriented NW-SE and presents a beach–dune barrier formed on the top of a calcareous bedrock [23]. This latter forms some headlands and outcrops in several points over both the foreshore and the upper-middle shoreface [12]. The whole physiographic unit is supplied from three sources: shallow marine carbonate factories produce calcareous biogenic sand; rivers that flow north-central Apulia transport terrigenous silicate detritus and heavy black minerals of volcanic origin; coastal cliffs furnish silicate and carbonate clasts [16,24,25]. However, it is affected by erosion and shoreline retreat for almost half a century as a result of natural processes and human interferences [12,17,24]. The grain size ranges from fine to very fine for the silicate grains, and from medium to coarse for the carbonate component [12,25].

A marked shoreline recession trend of the Cesine Lagoon beach was noticed nearly 20 years ago [14,26]. Such a process threatened the ecological balance and biodiversity of the lagoon, which are both relevant [21,27]. Thus, attached (connected to the shoreline) and detached low-crested breakwaters were quickly built (Figures 1 and 2) in order to counteract the shoreline retreat [12]. However, since that time, the barrier system was affected by a continuous change in morphology. Figure 2 shows the trend in shoreline change from 1948 to 2010 at the middle sector of the Cesine Lagoon (for details see Delle Rose 2015 [12]). An apparent recession trend is visible at the central part, with probable acceleration from 1988 to 1998 (blue, yellow, and green lines in Figure 2). Then, breakwaters caused the re-shaping of shoreline, tombolos formation, and washover fan accretion (red line and bodies in Figure 2).

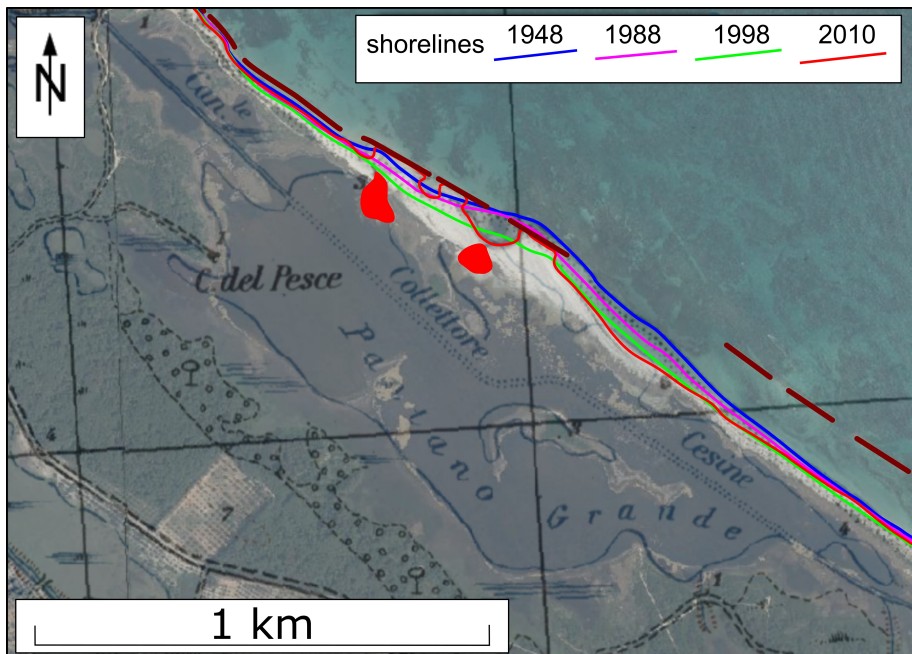

**Figure 2.** Changes of shoreline position (1948–2010) at the middle sector of the Cesine Lagoon (after [12], modified). Elaborated from the viewer service of the Italian National Geoportal [28]. Brown lines symbolize the breakwaters that were built in the early 2000s (see text). Large washover fans developed after the building of breakwaters are painted in red.

The southeastern Apulian beaches are subjected to an annual progradation/retrogradation cycle [12,22]. Their emerged sectors are usually wider, higher, and coarser in summer than in winter, when significant sand volumes are temporarily transported nearshore by storms to form submerged coastal bars, and meanwhile the dunes are eroded. From spring, the nearshore sand is transported back to the shoreline to form the onshore beach, and the finest sand fraction is moved inland by wind.

### 2.2. Wind and Wave Climates

The two main 6-month climatic seasons of October–March (cold–wet, CW) and April–September (warm–dry, WD) characterize the Mediterranean climate over the Apulia region [29,30]. Indeed, the southwards migration of the Atlantic storm track (causing mainly strong winds from the southern sector over the area) and the descending mid-atmospheric throughs from Northern Europe (often accompanied by northern cold outbreaks) are typical of the CW period, while good weather with scarce precipitations prevails in the rest of the year. The region is characterized by the predominance of northerly winds, although also winds from southeast are significant for frequency and strength [31]. The wave climate of the southern Adriatic Sea strictly reflects the above wind climate, being characterized by highest and longest waves coming from N and SE [32]. The wave climate intensity of the southern Adriatic moderately increased during the last decades. In fact, statistical trend analysis of the late 1970s–late 2010s wave time series detected a slight increase in the yearly mean values of significant wave high, wave power, and energy period [33]. The general surface current circulation of the Adriatic Sea is cyclonic, including a northwestern flow along the eastern side and a southeastern flow along the western side. Secondary gyres and coastal currents can seasonally prevail when the mean eddy kinetic energy is at a maximum in the southern Adriatic [34,35].

Measuring instruments installed in southeastern Apulia (see panel a in Figure 1) allowed the production of wave and wind data sets that are publicly available. The Monopoli buoy (108.5 km NW from the study area; mooring depth 85 m) belonging to the National Tidegauge Network (Rete Mareografica Nazionale, in Italian) is managed by the Higher Institute for Environmental Protection and Research (Istituto Superiore per la Protezione e la Ricerca Ambientale, ISPRA) [36]. It was in operation from 1989 to 2014 with several breaks due to failure or service intervals [37,38], and it started operating again in 2021. Fifty ($H_{s50}$) and 100 ($H_{s100}$) year return heights were estimated in $6.9 \pm 0.8$ and $7.4 \pm 0.9$ m, respectively [39]. The wind gauge stations reported in Figure 1a are the closest to Cesine on the Adriatic coast and belong to different authorities. The stations placed at Brindisi, Cesine, and Otranto are managed by the Apulia Civil Protection. The first has been in operation since 2009, while the second and third only came into operation in July 2020. The station located at San Cataldo (Figure 1a) is part of the Apulia Meteomarine Network (Sistema Informativo Meteo Oceanografico delle coste Pugliesi, in Italian) and was in operation from 2007 to 2014 [40]. Finally, in the Otranto harbor, a wind gauge installed at the mareograph station of the National Tidegauge Network has been in operation since 2010 (hereinafter referred to as Otranto-ISPRA station) [36].

## 3. Materials and Methods

Despite the importance in coastal defense and environmental conservation of the barrier system of the Cesine Lagoon (Section 2), there are still no specific studies on its geomorphological dynamics and site-related wind–wave conditions. This makes, on one hand, hazard zoning unfeasible and, on the other hand, comparative studies (for example with other Mediterranean barriers) unattainable. The present study was undertaken to address, at least partially, these knowledge gaps. Materials and methods used for the meteorological and geomorphological investigations are exposed in Sections 3.1 and 3.2, respectively.

### 3.1. Meteorological Investigation

The wind conditions affecting the study area are inferred from data sets reported in the *Annali Idrologici*, which are published by the Apulian Civil Protection [41–47]. These data sets consist of daily averages together with four 30 min averages at four selected hours of the day (00, 06, 12, 18). Data from the Otranto-ISPRA station [36] are also processed and consist of 10 minute averaged wind data. The considered data range from 2015 to 2021 is in accordance with the availability of Google Earth high resolution images and the duration of geomorphological investigation (see below).

The Weibull distribution is used to approximate the frequency distributions of the wind speed. It is a widely used distribution, suitable for several environmental variables, including wind speed [48], for its characteristics of high versatility in shape and the fact that it allows analytical expressions for almost all of its relevant statistical features. Thus, the Weibull cumulative distribution $W$

$$W = 1 - exp[-(v/s)^k] \tag{1}$$

is used as a regression equation for the cumulative frequency distribution of the measured wind speed $v$. The maximum probability wind speed $Vm$ and the percentiles speeds $V(p)$, where $p$ is the cumulative probability to have $v > V(p)$, are expressed as a function of the regression parameters $s$ (scale parameter) and $k$ (shape parameter) as

$$Vm = s[(k-1)/k]^{1/k} \tag{2}$$

$$V(p) = s[ln(1/p)]^{1/k} \tag{3}$$

A coastal storm is a "meteorologically-induced disturbance to the local maritime conditions (i.e., waves and/or water levels) that has the potential to significantly alter the underlying morphology and expose the backshore to waves, currents and/or inundation" [49]. Moreover, in agreement with Boccotti that defines a sea storm "as a succession of sea states in which the significant wave height exceeds a fixed threshold for a duration of at least equal to 12 h" [50], this time interval is herein considered.

To obtain a significant value for the expected wave height during the statistically extreme wind events (coastal storms), the spectral peak offshore wave height $H$ is calculated from the offshore 10 m height wind speed $U$ of the Bolam–Moloch model maps archive [51], from which the fetch $F$ and duration of the oversea wind condition are also verified. The used equation is [52,53]

$$gH/U^2 = 0.0016(gF/U^2)^{1/2} \tag{4}$$

where $g$ is the gravity acceleration.

The storm surge is a typical increasing of the sea level because of the storm atmospheric pressure drop and other effects, such as the blocking by the coastal platform of the Ekman sea current caused by the surface wind stress [52,54]. The coastal surge $S$ in storm conditions can be estimated from the following empirical equation [52,53]:

$$S = S_0 F_s F_m \tag{5}$$

where $S_0$ is directly correlated to the pressure drop of the atmospheric pressure minimum and $F_s$ and $F_m$ are corrections depending, respectively, on the depth of the coastal platform (shoaling factor) and on the atmospheric storm motion (speed and direction of the pressure minimum with respect to the coastline). They can be calculated by empirical tables ([52] pp. 212–215) after estimating the pressure drop, speed and direction of the storm center (minimum sea level pressure center) from the Bolam–Moloch archive [53].

*3.2. Geomorphological Investigation*

The study of shoreline mobility and geomorphological processes acting on the beach–dune barrier of Cesine Lagoon was carried out using field surveys and analysis of high resolution Landsat 8 satellite images. Preliminary geological investigations began in 2005, a few years after the construction of the breakwaters [14]. At the end of September 2016, before the beginning of the autumn–winter erosion (retrogradation phase, see Section 2), topographic profiling and sediment sampling check of the larger beach facing the Cesine Lagoon were made according to literature procedures, respectively, based on spirit leveling along the selected profiles and mechanical sieving of sand and gravel [55,56]. Since that time, several field observations on the morphological changes affecting the northwest and central sectors of the barrier system have been made (occasionally coupled with morphology measurements), which have been instrumental in the analysis and interpretation of satellite images. As the southeastern sector (the stretch of the coast southeast of Ponte di Carlo, see Figure 1) was apparently stable, it has not been investigated so far.

Sets of middle (April 2010, June 2012) and fine spatial resolution images (July 2015, July 2017, July 2018, June 2020, and September 2021) were acquired from Google Earth Pro. Then, they were manually digitized in a Geographic Information System (GIS) software to determine the morphological changes that occurred in the study area. For such an aim, the use of baselines corresponding to fixed points is crucial. The visual interpretation was performed according to the rudiments of Geographic Object-Based Image Analysis (GEOBIA) [57,58]. As a consequence of on-site observations and measurements, dune ridges, washover fans, and gravel beaches are the geographic objects on which the satellite imagery interpretation was focused.

Special care was given to shoreline detection and shoreline change interpretation. As a matter of fact, because of the dynamic nature of the land–water interface, a functional definition of the "shoreline" was required [59]. Shorelines inferred from satellite images do not represent normal or average conditions but the land–water interface at one instant of time. Actually, they are "instantaneous shorelines" [60–63] (see Appendix A). Both the imagery interpretation and the field observation allowed to obtain insights on the littoral drift.

The dune vegetation line was used to detect changes in the horizontal position of the dune toe [64,65], while the corresponding break in slope (local maxima of curvature at the beach–dune intersection) was identified on the field to verify the reliability of the image interpretation. Finally, it must be noticed that the resolution of the 2010 Google Earth image resulted in being sufficient to draw shorelines and vegetation lines in some figures (see below).

## 4. Results

The achieved results are exposed in the following sections. The wind–wave conditions defined for the study area are first presented (Section 4.1). Wind speed percentile values, and minimum 1-year return times wind speeds resulted from statistical analysis. Significant wave heights and storm surge values for extreme events were obtained using Equations (4) and (5), respectively. The synoptic description of selected storms concludes the results of the meteorological investigations. Section 4.2 is focused on barrier profiling and sediment grain size check, while Section 4.3 faces the comparison among instantaneous shorelines, obtained from the Landsat 8 imagery, to infer the prevailing direction of the littoral drift. Finally, geomorphological changes that affected dune toes, washover fans, and gravel beaches during the considered 7 years are described in Section 4.4.

*4.1. Wind–Wave Conditions*

4.1.1. Wind Statistics

A wind data set for the considered 7 years in the Southeastern Apulia coast is available only from the Brindisi station. Cesine and Otranto have only 1 year available data set in the same period (2021) [41–47]. The Otranto-ISPRA station, despite having a suitable

position along the shore in the Otranto harbor, has an insufficient cover of about 3 years (2015, 2020, 2021 and only part of 2016 and 2019) [36]. Thus, for completeness and position, the choice of the Brindisi data set as a proxy for the Cesine coast was somehow obliged (see Figure 1a for location). Nevertheless, a detailed analysis was also performed, comparing the 2021 Brindisi data set with both that available from Cesine and that available from the Otranto-ISPRA station to evidence the extent and limits of this proxy (Figure A2). This is briefly discussed in Appendix B.

Figure 3 shows the 7 years total and seasonal wind roses from the Brindisi station, considering the two main 6-month climatic seasons (i.e., the cold–wet October–March and the warm–dry April–September) that characterize the Mediterranean climate (see Section 2.2). A seasonal relative frequency increase from the southern sector is evident in the CW season because of the increase in the incoming Atlantic storms over the Mediterranean region.

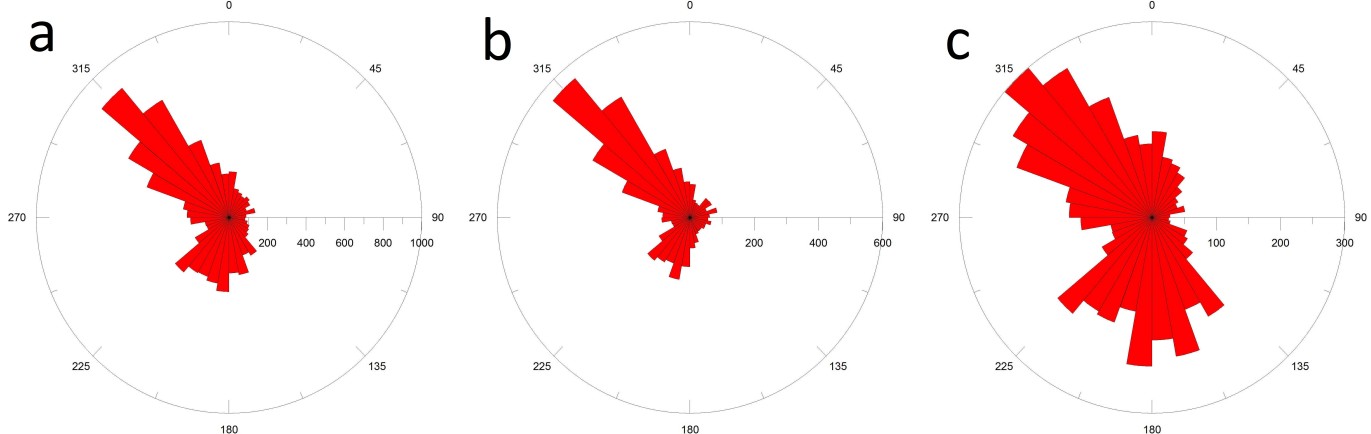

**Figure 3.** Wind roses for the 7 years data set of the Brindisi station (2015–2021); (**a**) all data; (**b**) warm–dry season; (**c**) cold–wet season.

A statistical analysis was made over the daily averages of the wind speed because they should be more relevant for prolonged coastal effects in general and, in particular, for the fact that a wind–wave intensity relation such as Equation (4) is valid for prolonged wind action over a long fetch that requires over 12 h of duration time [50,53]. To proceed with the statistical analysis of the proxy data set, two different 90° sectors that cover the incoming winds from offshore of the Cesine coast (315° NW–45° NE and 45° NE–135° SE) were selected (Figure 1). This choice aims to take into account both the statistically prevailing directions (spread around the N-NW and S-SW axes, with less contribution from the E direction) and the main axis of the Cesine coast, along the NW-SE direction (see also the Brindisi and Cesine wind roses in Appendix B). These two sectors select the main meteorological contributions to strong winds (storms from Mediterranean Lows and Balkan cold outbreaks from Northeastern Europe throughs: see also Section 4.1.2), with direction components against (or, at most, parallel to) the coastline. An additional south sector (135° SE–225° SW) was also selected because of the importance of the Mediterranean storms associated to S-SW winds, although the effects of winds from this sector are expected to be quite small on the Cesine coastline. The remaining western sector was disregarded as being considered irrelevant for the purpose of this work because of the coastal orientation and also the presence of a pine forest west and northward of the Cesine Lagoon.

The statistical analysis of the 7-year data set of wind speed from the Brindisi station was performed by both sector and season. The cumulative frequency distribution of the wind speed was studied by regressions with the Weibull probability cumulative distribution (Section 3). The regressions were performed for both the whole frequency distributions to obtain the bulk distribution parameters and the wind speed percentile values, and the upper tail only (starting from the inflection point of the cumulative frequency distribution, see also Appendix B) to obtain the minimum 1-year return times wind speeds. The 1-year return time was selected to characterize the most extreme events recorded with some

possible periodicity and strong coastal effects in the 7-year data set. The results are shown in Table 1 for the percentile values of the measured wind speed, and the daily wind speed interval (minimum–maximum) found in the Brindisi data set for speeds with 1-year minimum return times. The last two columns also show the maximum–minimum wind speed interval at 10 m over the sea surface modeled by the Bolam–Moloch model for the corresponding minimum 1-year return time selected events (daily wind averages), and the related maximum–minimum range significant offshore wave height for the same events, calculated by Equation (4).

**Table 1.** Features of the 7-year probability distributions for the wind speed in the Brindisi station by period of the year (CW: cold–wet, WD: warm–dry) and direction sector. N: 315° N–45° NE, E: 45° NE–135° SE, S: 135° SE–225° SW. *Vm*: maximum probability wind speed. *V(p)*: *p*–percentiles for *v* > *V(p)*. 1Y *v*(min–max): maximum and minimum wind speed in the Brindisi station, found for the events with minimum 1 year return time. 1Y *U*(min–max): same as above but for the Bolam model oversea wind speed. 1Y *H*(min–max): same as above but for the calculated maximum spectral wave heights. Speeds in m/s, heights in m.

| Season/Sector | *Vm* | *V*(0.25) | *V*(0.50) | *V*(0.75) | 1Y *v* (min–max) | 1Y *U* (min–max) | 1Y *H* (min–max) |
|---|---|---|---|---|---|---|---|
| CW-N | 0.8 | 3.7 | 2.1 | 1.0 | 9.0–11.6 | 16–20 | 3.7–5.6 |
| CW-E | 1.6 | 2.8 | 2.0 | 1.2 | 5.6–6.2 | 15–17 | 3.4–4.2 |
| CW-S | 2.9 | 3.8 | 3.0 | 2.2 | 6.0–6.8 | 16–20 | 5.8–7.2 |
| WD-N | 2.3 | 3.7 | 2.7 | 1.7 | 7.0–7.7 | 12–15 | 3.4–4.8 |
| WD-E | 2.0 | 2.5 | 2.0 | 1.5 | 4.2–4.3 | 10–10 | 2.3–3.2 |
| WD-S | 3.3 | 4.0 | 3.3 | 2.6 | 5.0–5.6 | 8–15 | 2.4–5.4 |

Thus, Table 1 shows some main characteristics found for the coastal wind conditions in the considered 7 years. First, somehow unexpectedly, the WD season appears to be more 'windy' than the CW one for what concerns the bulk properties of the statistical distribution (*Vm* and percentiles). This is a possible effect of the much more effective coastal breezes during the WD season. Moreover, examining the N and E sectors that are the most effective in shaping the barrier system of Cesine Lagoon, it appears that winds from the N sector are generally more intense. In addition, the N statistics in the CW season shows a very long tail toward high wind speeds (smaller *Vm*), suggesting a stronger contribution of local storms. This is confirmed by the wind speed ranges for the minimum 1 year return time that show the presence of some very strong events in the CW season from the northern sector. For these events, the wind speed at 10 m oversea is between 16 and 20 m/s, which cause the highest waves impinging over the coast (excluding winds from the S sector, possibly causing offshore waves only). The overall range of significant (offshore) wave height for the extreme events (minimum 1-year return time) from both the N and E sector is between 2.3 and 5.6 m, with increasing height in the CW season. Higher wave heights are calculated from the S sector, mainly because of the longer fetch potentially available, but, in spite of their documented disrupting effect over the Ionian Apulian coast [53], they have minor effects over this part of the Adriatic coast because of its opposite orientation.

### 4.1.2. Coastal Storms

This section concerns a brief synoptic description of some significant events selected from the minimum 1-year return time events found in the Brindisi data set to outline the synoptic conditions that trigger strong wind events over the considered coastal area. The number of events were found to be about half-dozen per direction sector, as expected for 1-year return time in a 7-year data set. Figure 4a,b show the wind field at 10 m and the geopotential height at 500 hPa for the strongest wind event recorded at the Brindisi station, in February 2020, with over 11 m/s wind speed from the N sector. As shown in these figures, this event was triggered by a deep incoming through from the North-Eastern Europe, intensifying over the Balkan region, a typical condition that advects cold northerly winds over the Adriatic sea. The significant offshore wave was found to be over 5 m

height (Equation (4)) and in this case, Equation (5) gives an estimate of about 1.2 m for the storm surge.

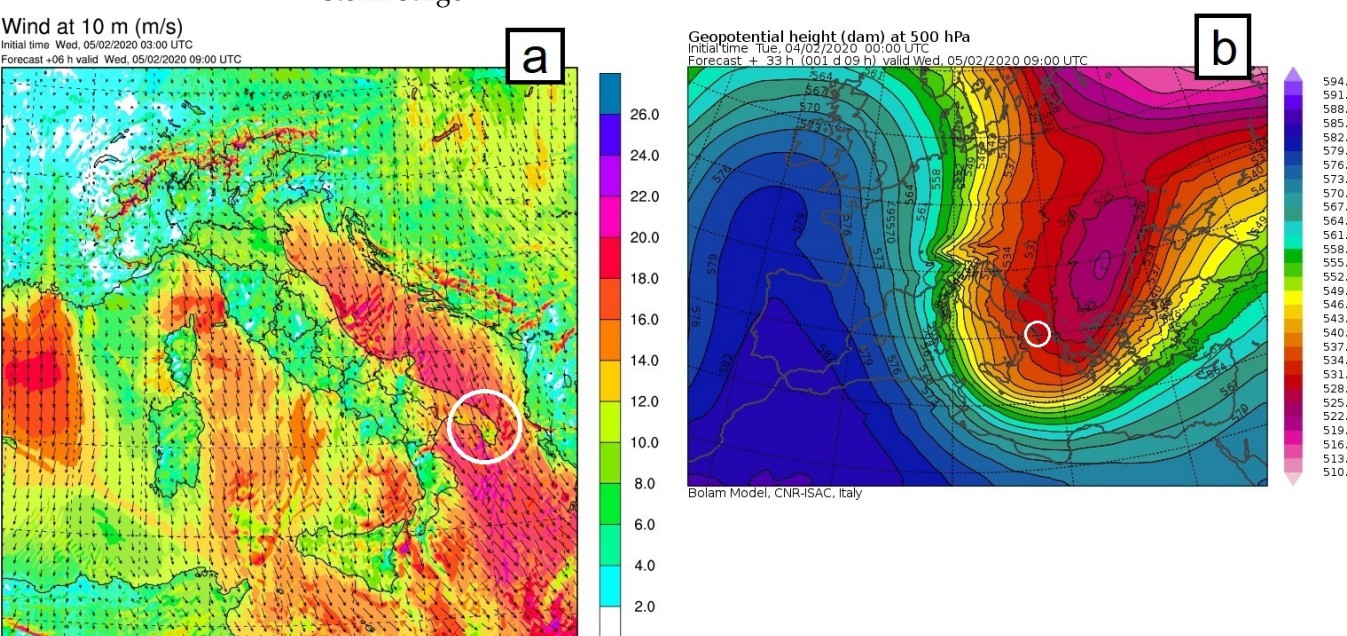

**Figure 4.** Windfield at 10 m height in m/s (**a**) and geopotential height in at 500 hPa (**b**) from the Bolam–Moloch model archive, for the wind storm of February 2020. White circles mark the Southern Apulia region.

Figure 5a,b show the wind field and the geopotential height for the Eastern outbreak of March 2015. The triggering event was a Mediterranean low but still generated by an incoming through from Northeastern Europe. In this case, however, its intensification over the central Mediterranean area causes a different position of the low pressure center and a different condition for the triggered Adriatic winds, that assume a more eastern direction. The storm surge estimated by Equation (5) is of about 0.4 m height, but the direction perpendicular to the coastline of easterly sea waves of over 3 m height (Figure 5 and Table 1) can enhance the total flooding effect over the exposed coast in this case.

For completeness, the last event was selected from the southern sector. This very strong event of October 2018 is well known and studied in the scientific literature, where it was named the 'Vaia' storm for the damage and disruption caused mainly in the Northeastern Italian regions but also over the Ionian Apulian coast (Delle Rose et al. [53] and cited bibliography). As Figure 6a,b show, in this case, the strong wind field is southerly oriented in the Ionian sea, and this is related to the place of pressure minimum, located now in a more northwestern position as generated from a through located northwestern of the Mediterranean, over the Atlantic Ocean. This different generation/position of the minimum is also responsible of the southerly wind speed orientation over the Adriatic sea and Ionian sea, with a typically longer fetch. The calculated significant wave is about 6 m height, and the storm surge can be estimated at about 1.2 m in height.

Finally, a diurnal tide contribution of about 0.3 m is also possible to increase the coastal surge for storms in this site (Section 2). All the above considerations show that an increase in the sea level of the order of 1 m can be expected along the considered coastline for stormy events with a return time of about 1 year.

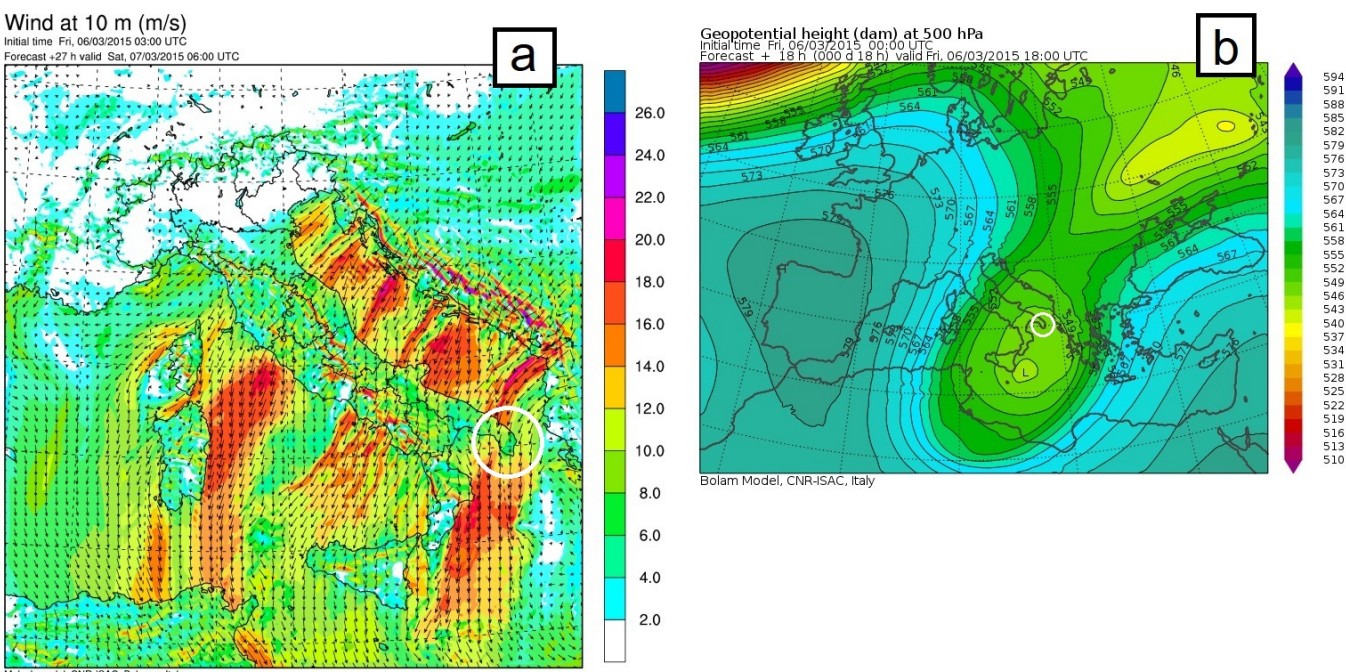

**Figure 5.** Windfield at 10 m height in m/s (**a**) and geopotential height in at 500 hPa (**b**) from the Bolam–Moloch model archive, for the eastern outbreak of March 2015. White circles mark the Southern Apulia region.

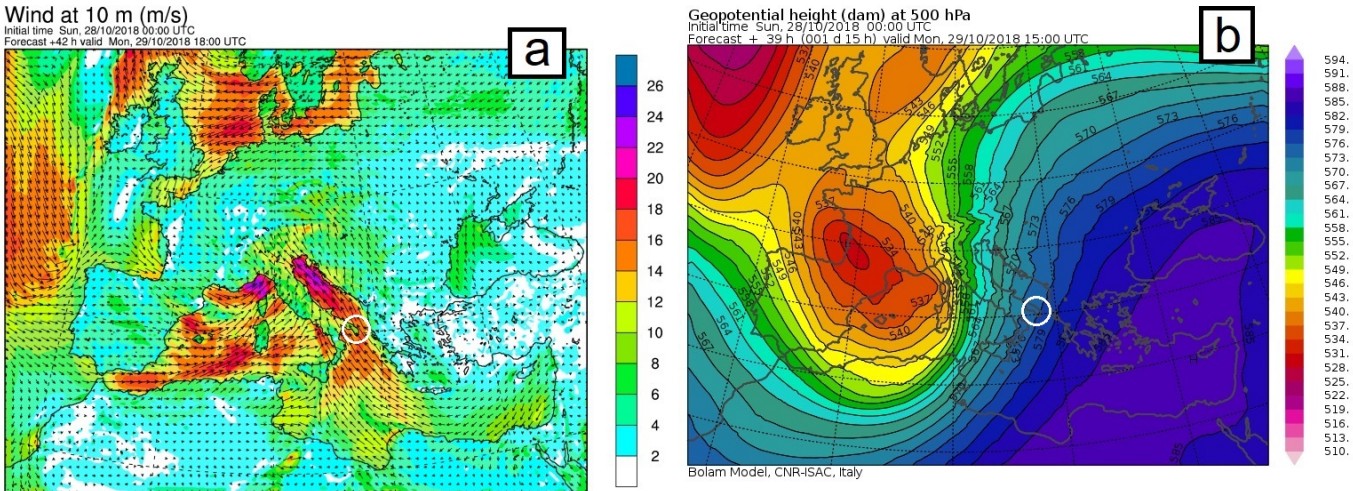

**Figure 6.** Windfield at 10 m height in m/s (**a**) and geopotential height in at 500 hPa (**b**) from the Bolam–Moloch model archive, for the 'Vaia' storm of October 2018. Note the long southerly fetch. White circles mark the Southern Apulia region.

*4.2. Barrier Profiling and Sediment Grain Size Check*

On 27 September 2016, as a result of the specific field investigation, three cross-barrier profiles were measured and sampled in the northwest sector of the Cesine Lagoon, where the barrier system was larger than in the central sector of the study area (Figure 7). The aims were (1) to sketch the shape of the upper-shoreface/foreshore/backshore and foredune/backdune; and (2) to check the grain size class distribution.

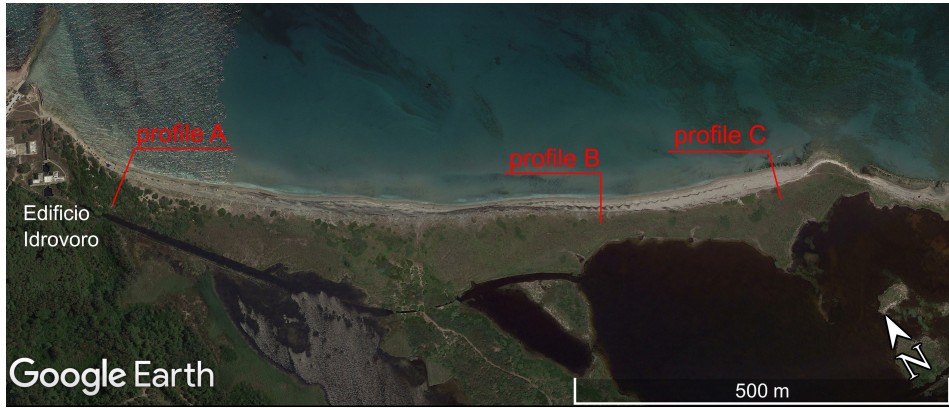

**Figure 7.** Selected profiles for sediment grain size check (see text). Background image is dated 19 July 2015 (40°21′53.74″–40°22′30.29″ N, 18°18′45.53″–18°19′45.96″ E; eye elevation of 1.1 km).

The beach was narrower and flatter at the NW side (profile A) than at the SE side (profiles B, C), thus suggesting the prevalence of southeastern-ward longshore transport. Well-shaped storm berm over the backshore and break-point step at the base of the foreshore were observed along the southeastern profiles (Figure 8). The average foreshore slope was about 1/5.

Differently from the beach, the dune was wider on the NW side. However, on the SE side, the dune was higher (Figure 8). Such features were not the result of short-term morphodynamic change but rather of multi-decennial interaction between physical and vegetation factors. Sparse plants of *Elytrigia juncea* (L. Nevski) characterize the fore dune and the crest area, while the more internal zone of the backdune show *Spartina versicolor* (E. Fabre) and Mediterranean shrubs. The presence of this vegetation type suggests a progressing stabilization of the backdune. In the following five years, this environment/landscape setting will not apparently change, except for a moderate retreat of the dune toe at the NW side (see Section 4.4.1).

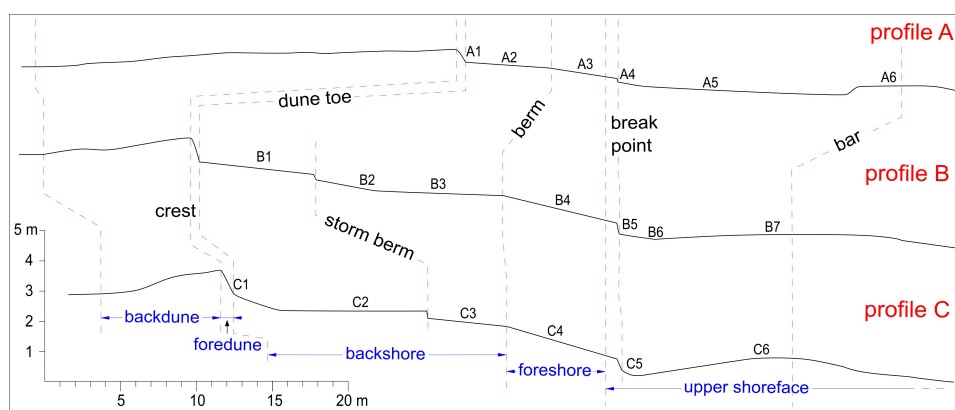

**Figure 8.** Barrier profiling and sampling (vertical scale is twice the horizontal one).

Grain size decreased both seaward and landward from the base of the swash zone, changing from very coarse sand (mixed with gravel at profile A) to fine sand. The backshore and foreshore had a similar grain size (coarse-medium sand), except for profile B with the finer samples B1 and B2 (medium-fine sand; see Appendix C for details).

### 4.3. Instantaneous Shorelines Comparison

For 2015–2021, the comparison of instantaneous shorelines digitized from Google Earth images suggests a net southeastward littoral drift trend. The shoreline retreated at the NW sides of each sector, and advanced at the SE sides (Figures 9 and 10). This is in agreement with field observations and measurements (see Figure A6 in Appendix D, an example of a

beach profile measured on site several times). However, during the CW seasons, temporary inverse transport was observed on site in conjunction with prevailing southeasterly winds (cf. Figure 3c). Moreover, although field observations were not systematically carried out during the 7 years of research, accretion and erosion dynamics were observed at the updrift and downdrift sides, respectively, of some coastal defense structures (especially in correspondence of the groyne located at the northwest end of the northwest sector; see Figure 9 for the location of this structure).

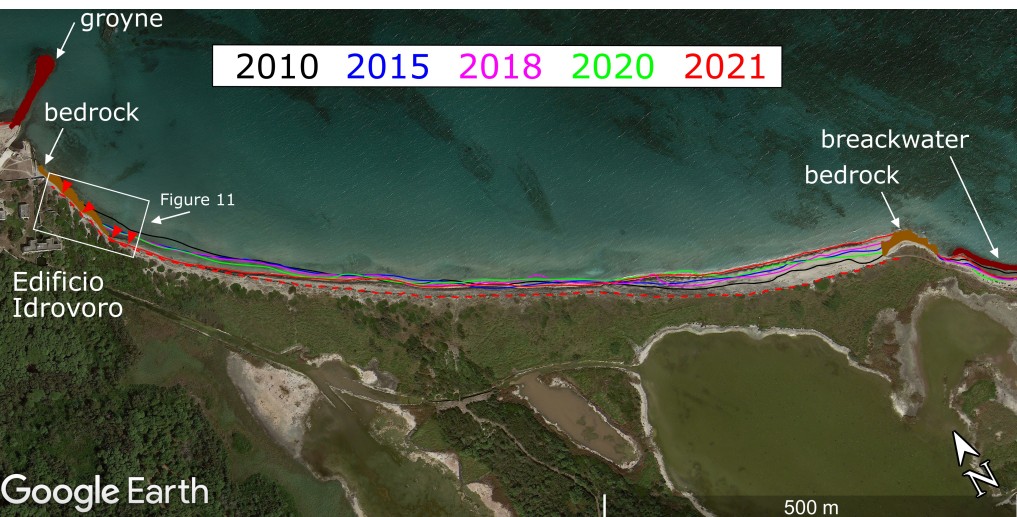

**Figure 9.** Shoreline position changes at the north-west sector (40°21′55.68″–40°22′31.60″ N, 18°18′45.90″–18°19′48.29″ E; eye elevation of 1.14 km). The 2010 shoreline is drawn as a reference. Dotted red line marks the 2021 dune toe; background image is dated September 2021.

With regard to the cross-shore dynamic, winter storms caused the steepening of the emerged beach, likely mobilizing significant volumes of sand seaward. Instead, a calmer summer wave climate pushed sand back onto the shoreline, widening and lowering the foreshore profile. This process was particularly apparent along the stretches of coast which are not protected by breakwaters, such as the one named "Foce della Conca della Corva" (Figure 10).

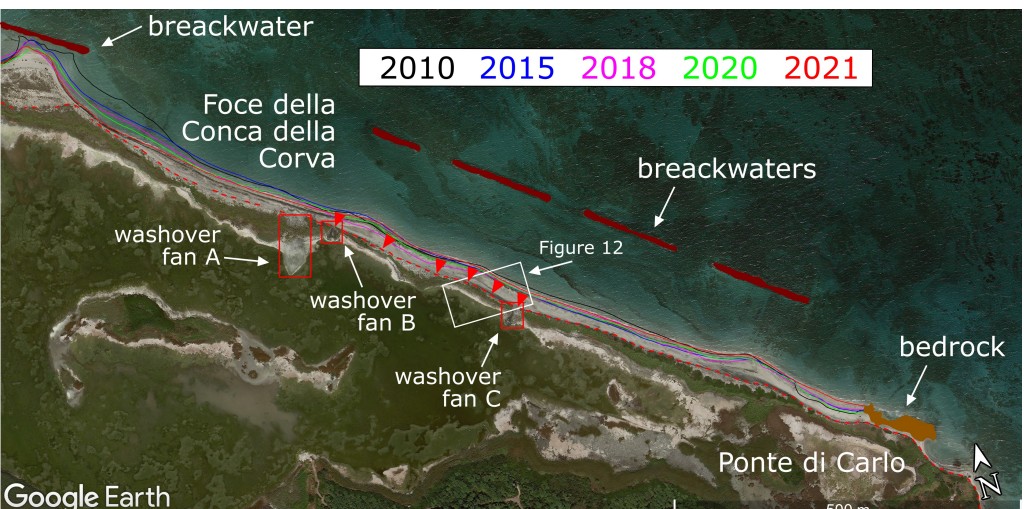

**Figure 10.** Shoreline position changes at the central sector (40°21′05.90″–40°21′41.04″ N, 18°20′06.04″–18°21′14.75″ E; eye elevation of 1.28 km). The 2010 shoreline is drawn as a reference. Dotted red line marks the 2021 dune toe; background image is dated September 2021.

*4.4. Geomorphological Changes*

4.4.1. Dune Toe Retreat

Throughout the Cesine barrier system, the dune height rarely exceeds 3 m. At the northwest sector, the dune ridge is even lower than 1 m (Figures 8 and 11).

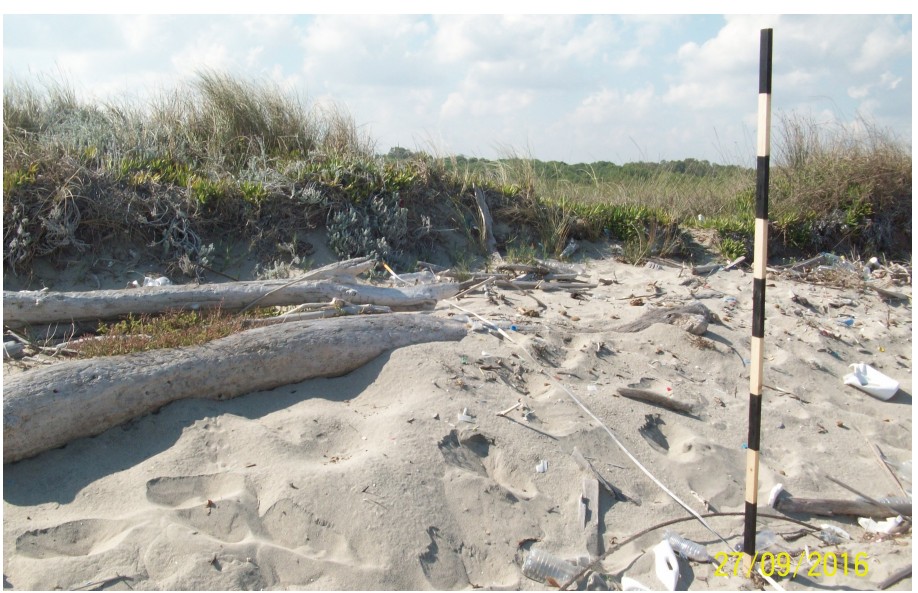

**Figure 11.** Measurements of dune width and height (40°22′06.85″ N, 18°19′30.30″ E) 27 September 2016; graduated rod, placed at the dune toe, is subdivided in 20 cm-long segments. The height of the dune ranges here from 0.9 to 1.1 m, the maximum value found in the northwest sector. The distance between dune toe and vegetation line is 2 m.

As observed on site, since at least the 2019–2020 CW season, this sector has been affected by dune toe retreat due to the erosion action of the swash (Appendix D, Figures A5 and A6). As a matter of fact, at the end of 2021 WD season, the beach almost completely disappeared at the northwest side of the northwest sector, and thus the bedrock cropped out along the land–water interface (Figure 12).

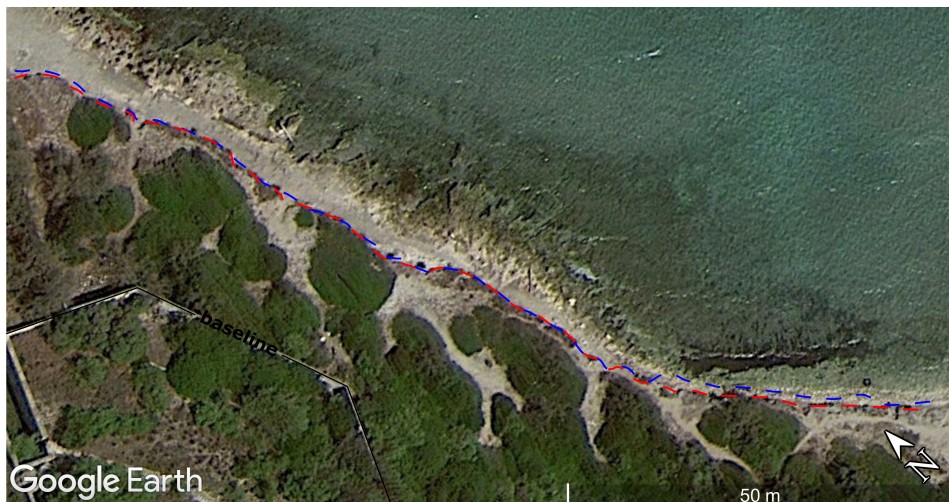

**Figure 12.** 2015–2021 dune toe retreat at the northwest side of the northwest sector (40°22′20.84″–40°22′25.14″ N, 18°18′57.41″–18°19′00.27″ E; eye elevation of 107 m). Blue dotted line and red dotted line mark the positions carried out from July 2015 and September 2021 satellite images, respectively; background image is dated September 2021.

The retreat of the boundary between the beach and the eolian deposits was also detected in some places of the central sector. As an example, Figure 13 documented an about 5 m dune toe retreat along more than 20 m of the barrier system. Despite the short observation period, it is quite significant. Unlike the case of Figure 12, the one of Figure 13 was likely due to wave run-up during storm events as deduced from field investigations.

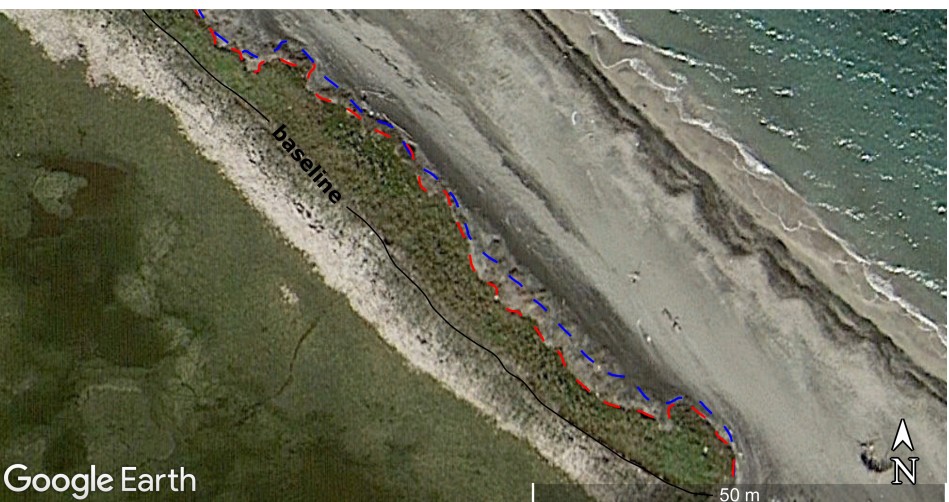

**Figure 13.** 2015–2021 dune toe retreat at the central stretch of the central sector (40°21′21.03″–40°21′22.94″ N, 18°20′37.03″–18°20′41.84″ E; eye elevation of 111 m). Blue dotted line and red dotted line mark the positions carried out from July 2015 and September 2021 satellite images, respectively; background image is dated September 2021.

The distance between the dune toe (break in slope) and vegetation on the crest of the dune was repeatedly measured during the field investigation. It was never greater than 2 m as in the case of Figure 11.

### 4.4.2. Washover Fan Accretion

The barrier system of the Cesine Lagoon has been affected by overwash events and accretion of washover fans since at least the building of the breakwaters in the early 2000s (Section 2). These processes occurred repeatedly during the investigated years as well, especially throughout the central sector of the barrier. The larger depositional body is the one named washover fan C in Figure 10. It has accreted since 2015 after the breaking of the dune ridge and tripled in size between 2018 and 2021, from about 150 to 500 $m^2$ (Figure 14).

In March 2023, the average elevation of washover fan C at low tide was about 0.4 m, while the length of the axes did not change in comparison with the one of September 2021 (Figure 15). Thus, more than 200 $m^3$ of sand was involved in its accretion, which is a significant amount for the studied barrier. Additionally, the little smaller washover fan B (Figure 10) experienced a similar development (Appendix D, Figure A8).

Both the washover fan A (Figure 10) and the washover fans located beyond the southeast end of the central sector (outside the investigated area, Figure 16) experienced different evolutionary stages. For these sedimentary bodies, the sand transport processes across the beach–dune barrier began between 2010 and 2012. The stabilization of the lagoon shores occurred between 2015 and 2017, while few changes were observed in subsequent years (see green and red lines in Figure 16).

### 4.4.3. Gravel Beach Formation

In the last few years, the formation of gravel beaches was observed along the stretch of coast, where the breakwaters are attached to the shoreline (Figure 1). This is a quite new

geomorphological process at the barrier system of the Cesine Lagoon. It is particularly intense in correspondence of the breakwater openings (Figure 17).

Gravels were produced from bedrock surfaces and breakwater stones, likely during the major storms. The deposits of gravel form foreshores that are steeper than sandy ones (sloped about 35°). However, during the WD seasons, they are in large part covered by eolian fine and very fine sand (Appendix D, Figure A9), to be exposed again during the CW seasons.

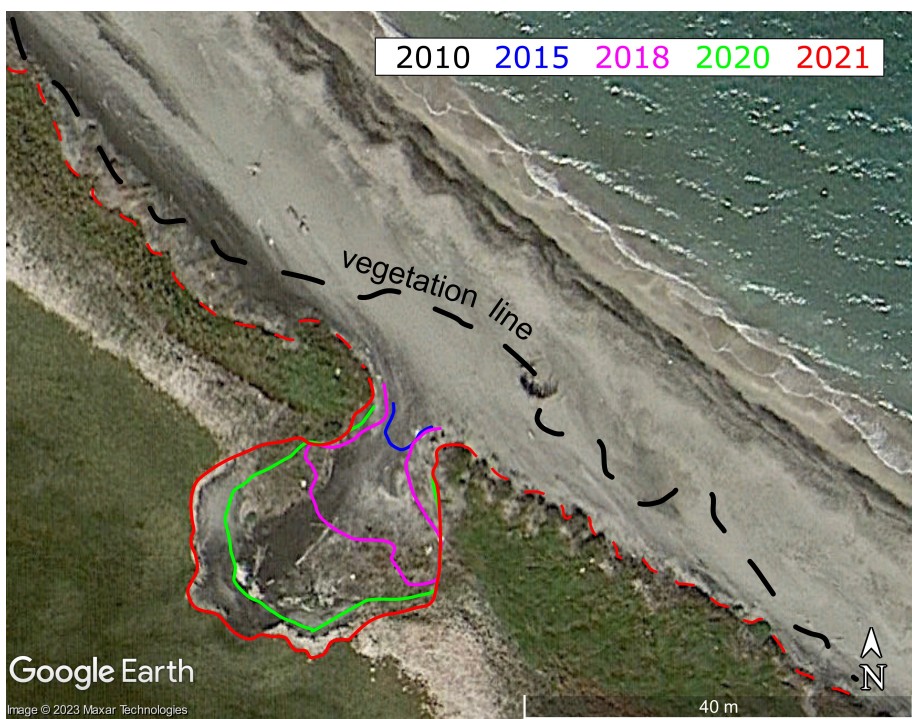

**Figure 14.** 2015–2021 washover fan C evolution. The 2021 dune toe in marked by red dotted line. Background image is dated September 2021 (40°21′20.05″–40°21′21.98″ N, 18°20′39.13″–18°20′43.11″ E; eye elevation of 81 m).

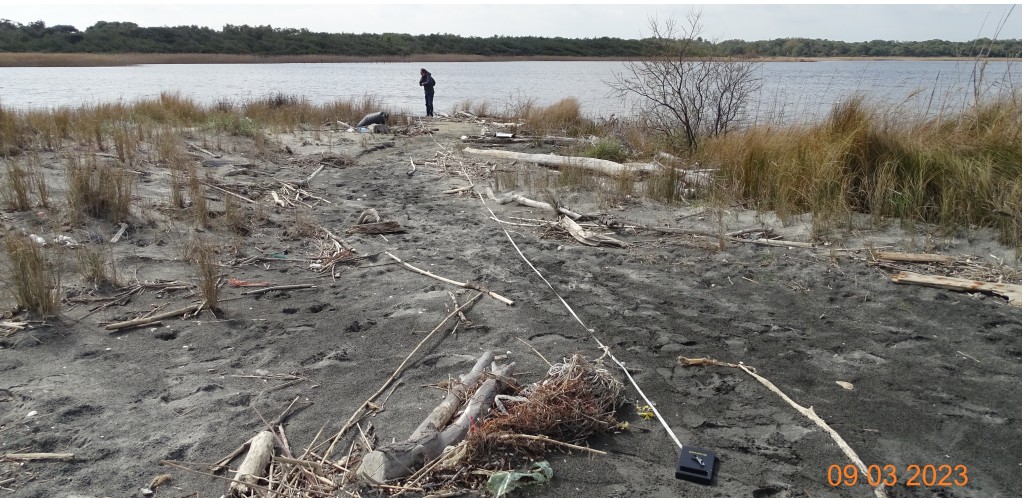

**Figure 15.** Measurement of the size of the washover fan C. A thin eolian deposit of very fine black sediments covers the seaside of the fan.

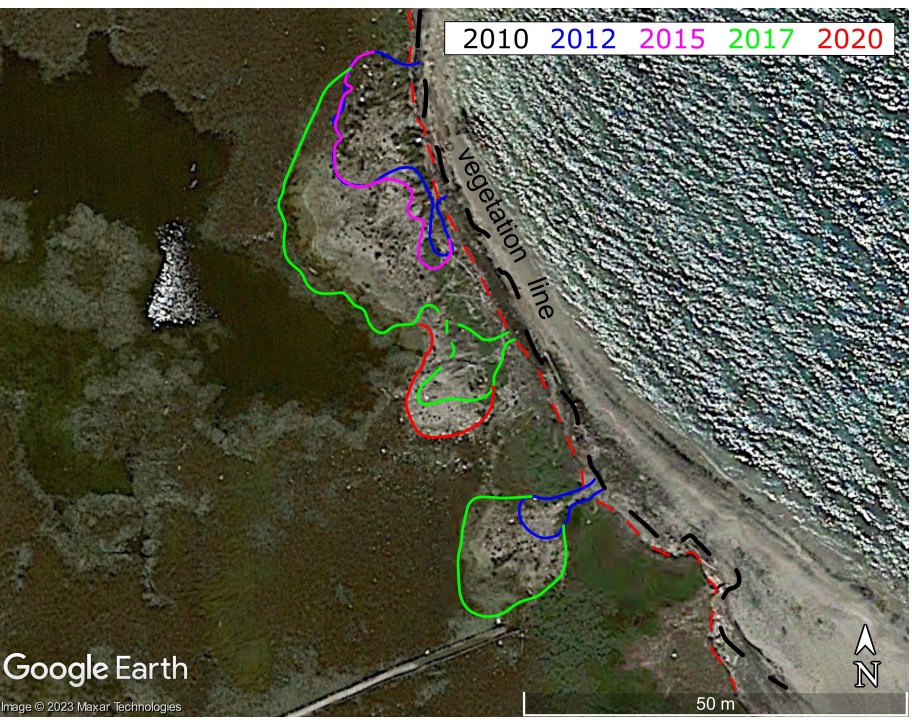

**Figure 16.** 2012–2020 evolution of the washover fans south of Ponte di Carlo. 2020 dune toe in marked by red dotted line. Background image is dated June 2020 (40°21′03.08″–40°21′05.70″ N, 18°21′02.39″–18°21′07.60″ E; eye elevation of 102 m).

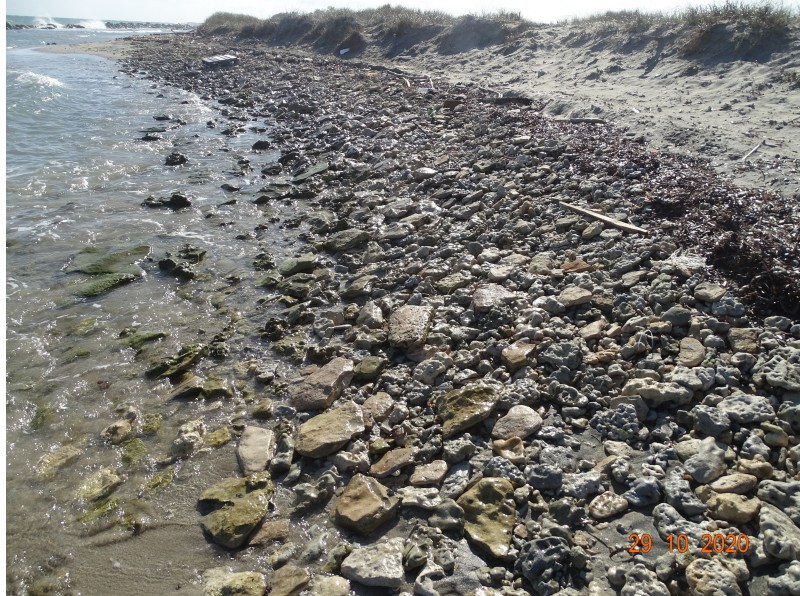

**Figure 17.** A gravel beach (40°22′02.90″ N, 18°19′37.85″ E) at the beginning of the 2020–2021 CW season.

## 5. Discussion and Conclusions

Wind–wave conditions are the main factor controlling the geomorphological shaping of the studied barrier. The contribution of other natural and anthropic factors is substantially negligible in the considered short term time. In fact, if on one hand, the tectonic subsidence and changes in sea level and sediment supply can produce significant effects only on medium-long terms, then on the other hand, no new human-induced process started after the building of the breakwaters in the early 2000s [12,17,24]. The definition of the weather conditions that affect the study area is consequently crucial in a climatological perspective.

Due to the lack of a wind data set covering 2015–2021 at the Cesine coast, a proxy was selected among the closest stations to infer the wind conditions of the study area (Section 4.1.1 and Appendix B). The performed statistical analysis shows that winds from N are generally more intense than winds from the east. Additionally, the north statistics in the CW season shows a very long tail toward high wind speeds, suggesting a stronger contribution of local storms. The overall range of significant wave height for the extreme events (minimum 1-year return time) from both the north and east sectors is between 2.3 and 5.6 m, with increasing heights in the CW season. Moreover, a storm surge of the order of 1 m can be expected for events with return times of about 1 year (Section 4.1.2). The above wind–wave conditions are comparable with the ones reported in previous studies [31,32]. Considering the flat beach profile and the large amount of middle to very fine sands on the foreshore and backshore (Section 4.2, Appendix C), significant shoreline mobility and geomorphological change can be expected at the beach–dune barrier of the Cesine Lagoon. Furthermore, since the dune crest does not exceed 1 m in height for long stretches of the barrier, vulnerability to flooding can be related to storms of the yearly return time (Section 4.1.2).

A net longshore transport of the beach sand from the northwest to southeast throughout the considered 7 years is inferred by satellite images analysis and on-site observations (Section 4.3). It is consistent with the prevalent northerly winds. Nevertheless, such a result is in good agreement with previous hydrodynamic conditions reported in the Apulian coast literature [16,24]. However, it must be noted a possible littoral drift imbalance. In fact, the output of sand due to littoral drift beyond the southeast end of the central sector seems to be not compensated by the input of sand through the NW end of the barrier. The disappearance of the emerged beach and the consequent cropping out of the bedrock at the NW end of the studied barrier system (Section 4.4.1) may be an evidence of this trend, which is likely caused by the building of groynes and breakwaters along the northern coast as previously forecasted by the authors (see [12,24]).

Dune toe retreat has been observed at different stretches of the studied coast. Where the backshore was eroded, it was due to the swash of the waves over the foreshore (both in normal and severe wind conditions), while in the presence of the emerged beach, to the wave run-up during storms, as a significant wave height can reach several meters for yearly return time events (Section 4.4.1). The largest retreat can be estimated in about 5 m, which is quite significant considering that it has occurred in few years (see Figures 9 and 10). At least in the case of the washover fan C (see Figure 14), the dune toe retreat seems to have driven the breaking of the dune ridge and the consequent development of the sedimentary body.

The study area experienced several episodes of overwash that allowed the accretion of washover fans during 2015–2021 (Sections 4.3 and 4.4.2). On the other hand, the breaking of the dune ridge and the consequent lagoon flooding were expected geomorphological and hydrogeological processes [18]. However, the collected data do not allow to establish whether the breaking of the dune ridge and the accretion of the washover fan are due to a major storm or are the result of a gradual process, in which ordinary weather conditions are decisive. As shown by several studies, washover fans can accrete over multi-year time scales also in the absence of large storms and with a significant role of the eolian processes (see, for example, [8,66]). It is apparent that more frequent and detailed field observations are needed to better investigate the weather conditions that actually drove such a morpho-sedimentological process. Anyway, their occurrence is essential for sustaining a barrier system faced with the rising sea level and increasing wave climate intensity (Section 2). As a matter of fact, by moving sand landward and supplying washover fans, overwash and eolian processes increase the width of the barrier and, consequently, its resilience to sea-level rise and resistance to wave erosion [67].

Another geomorphological and sedimentological process detected along the barrier system of Cesine Lagoon is the development of gravel beaches (Section 4.4.3). It is particularly active in correspondence of the stretch of the coast protected by the attached breakwaters (see Figure 1), where coarse clasts are produced by the wave erosion of

bedrock surfaces and breakwater stones (Figures 17 and A9). As the importance of the gravel beaches is recognized for shore protection, the interest in quantitative studies on their geometry and transport rate is increasing [68]. A number of essential parameters will have to be measured on site to describe the state and development of the gravel beach as a physical system. They are clast size, shape, and roundness through time and space; packing, orientation and vertical/horizontal gradation; grain mobility; and roughness to flow and infiltration [69].

Notwithstanding the effect of extreme events not having yet been determined, the geomorphological data collected since 2015 as a whole suggest an erosional trend of the studied dune–beach barrier. This trend is consistent with the multi-decadal one previously described for the whole physiographic unit, to which the Cesine coast belongs [12,26]. The same trend is also shown by other barriers and beaches in the central Mediterranean, thus determining that the impact of climate change on this process is a main research challenge [13,70]. For this purpose, it is essential to calculate the sedimentary budget [1–3]. Coastal geomorphological and sedimentological monitoring, including also the whole upper shoreface (for the case study the annual closure depth is about 5.5 m, see Section 2) is the most effective tool to achieve the goal [11,71]. In particular, to have real data on the geomorphological influence of extreme events, it is necessary to perform barrier profiling and sampling before and after storms. Moreover, for a deeper understanding of the relationships between geomorphological changes and wind–wave conditions, the distinction between along-shore and cross-shore processes is crucial in the next stages of the research [72,73]. In this view, the discussed results may be considered starting points for further data collection and analysis to assess the impact of climate changes and the threatening hazards on the lagoon barrier.

**Author Contributions:** Conceptualization, methodology, investigation, data curation, writing, and editing, M.D.R. and P.M. All authors have read and agreed to the published version of the manuscript.

**Funding:** This research received no external funding.

**Institutional Review Board Statement:** Not applicable.

**Informed Consent Statement:** Not applicable.

**Data Availability Statement:** Data supporting reported results can be found at the following links: http://www.pcn.minambiente.it/mattm/en/ (accessed on 13 February 2023), for the elaboration of Figure 2; https://protezionecivile.puglia.it/annali-e-dati-idrologici-elaborati (accessed on 30 December 2022), for the *Annali Idrologici* (Hydrological Annals), see Sections 3 and 4; https://www.mareografico.it/ (accessed on 20 February 2023), for the data of Otranto-ISPRA station; see Sections 3 and 4; http://www.isac.cnr.it/dinamica/projects/forecasts/index.html (accessed on 28 March 2023, for the calculation of wave data; see Sections 3 and 4.

**Acknowledgments:** Luca Orlanducci for the collaboration during geomorphological survey; Anna Lisa Signore for the technical assistance during beach profiling and sampling.

**Conflicts of Interest:** The authors declare no conflict of interest.

**Sample Availability:** Samples of the sand are available from the authors.

## Appendix A. Notes on Images Interpretation

Due to the nature of the coastal land–water interface, the shoreline position continually changes through time [59]. Based on the guideline of geographic-object analysis [57,58], time series of instantaneous shoreline extracted from freely available high-resolution (up to less than 0.5 m) satellite images are increasingly used in the reconstruction of geomorphological trends [60–63,74,75].

All the 2015–2021 satellite images used in this paper were taken during the WD season (Section 3) with calm wind–wave conditions and a flat or nearly flat sea surface, thus allowing easy image interpretations (Figure A1). Moreover, the daily variations of the shoreline due to the tidal fluctuations (about 1.5 m in plan as inferred from the measured

foreshore slope of Figure 8 and on-site observations) are comparable with the resolution of the images. Considering the scale of the Figures 9 and 10, the daily change of shoreline position due to the tides is smaller than the thickness of the line symbols. Therefore, the time at which the satellite images were taken does not affect the result.

A further consideration must be made as regards the digitized instantaneous shorelines. Taking into account the annual progradation/retrogradation cycle of the sand beaches of the study area, the width of the backshore tends to increase during the WD season (Section 2). Unlike the 2015–2020 satellite images that were taken between June and July, the 2021 image was taken in September (Section 3). Consequently, for 2021, the actual effect of the southeastward longshore transport (see Figure A1) may be exaggerated by the progradation process.

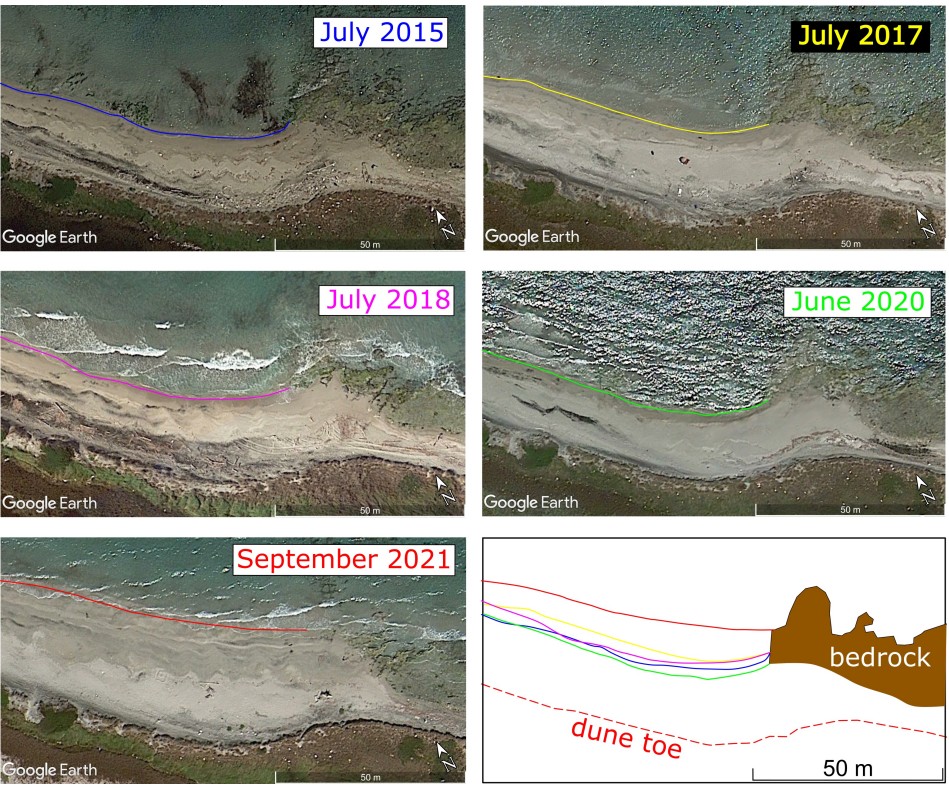

**Figure A1.** Instantaneous shoreline positioning at the SE side of the central sector (cf. Figure 10). Note that neither shoaling nor swash make it difficult to recognize the land–water interface (40°21′12.30″–40°21′14.18″ N, 18°20′55.54″–18°21′00.22″ E; eye elevation of 106 m).

To check the overlay of satellite images, some baselines are used. As an example, the artificial channel crossing the Pantano Grande and other minor water bodies (see Figure 1) allowed to check the georeferencing accuracy of Google Earth images used to make the Figures 9 and 10. Again, a fence wall was used for Figure 12 and the high water level in the lagoon, as marked by vegetation, for Figure 13.

## Appendix B. Wind Rose and Speed Distribution

Figure A2a–c show the wind roses for the year 2021 in Brindisi, Cesine and Otranto-ISPRA stations, respectively. Besides the commonly prevailing northern and southern directions, local differences are evident, although some of them can be ascribed to the position of the measurement site, that is semi-urban in Brindisi, on the shoreline, but sheltered by the port structure in the S-SE direction in Otranto, and behind a pine forest spreading mainly in the northern direction at Cesine. The sheltering of the Cesine station could be responsible for the less prevalent winds from the NW direction if compared with Brindisi and Otranto (Cesine is in between), while the S-SE sheltering of the Otranto station

could enhance the SW direction in comparison. The Brindisi station should have no evident prevailing sheltering from the coastal sectors but an almost uniform decrease in the oversea wind speed because of the urban boundary layer effects. A more enhanced reduction could be expected from the western sector, which is wholly inland, but this sector is not considered in this study.

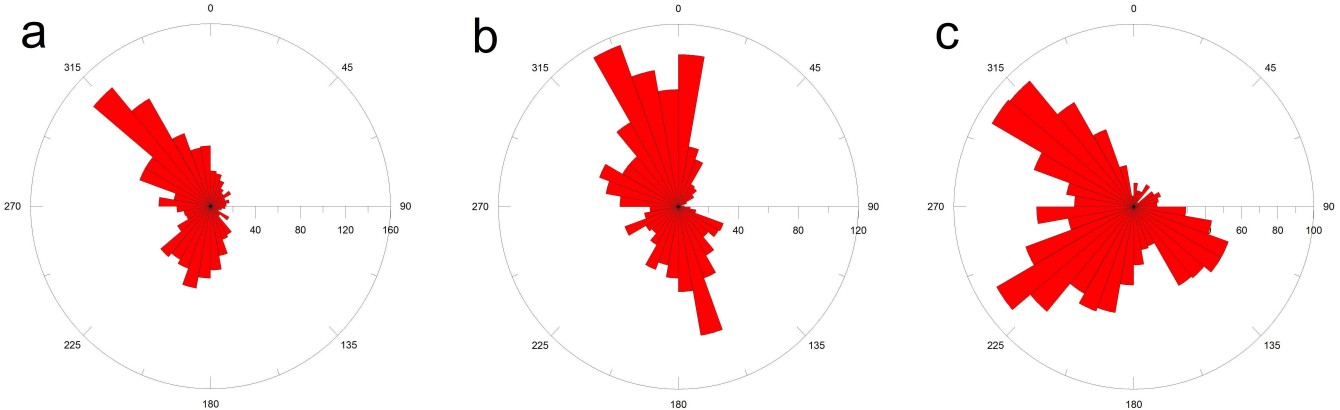

**Figure A2.** (**a**) Wind rose for the Brindisi station, year 2021; (**b**) wind rose for the Cesine station, year 2021; (**c**) wind rose for the Otranto-ISPRA station, year 2021.

Figure A3a–c show the 2021 wind speed distribution by direction sector for Brindisi, Cesine and Otranto. Brindisi and Cesine show good agreement for the wind speed distribution in the E and S sectors, while Otranto appears to have the maximum speeds in the N and E sectors but to underestimate in the S sector. In the N sector Cesine appears to underestimate the wind speed with respect to both Brindisi and Otranto, and the maximum speed is found in Otranto. These results agree with the hypotheses of sheltering from the N sector in Cesine, from the S sector in Otranto and a more uniform effect over directions in the Brindisi station.

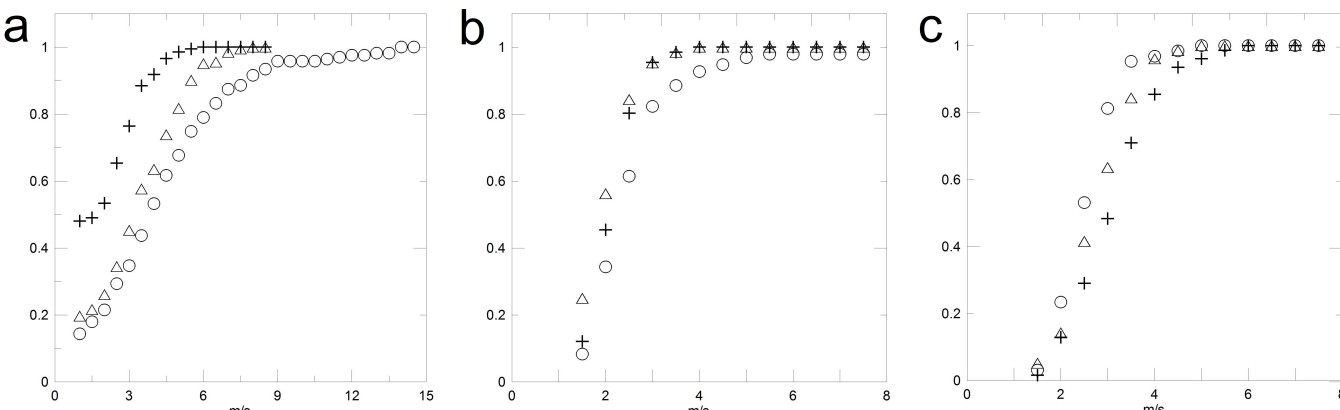

**Figure A3.** (**a**) Comparison of the cumulative probability (normalized frequency) distribution for the Brindisi station (triangles), Otranto-ISPRA station (circles) and Cesine station (crosses), N sector, year 2021; (**b**) comparison of the cumulative probability (normalized frequency) distribution for the Brindisi station (triangles), Otranto-ISPRA station (circles) and Cesine station (crosses), E sector, year 2021; (**c**) comparison of the cumulative probability (normalized frequency) distribution for the Brindisi station (triangles), Otranto-ISPRA station (circles) and Cesine station (crosses), S sector, year 2021.

**Appendix C. Grain Size Characterization**

Nineteen samples (Figure 8) were collected by grab sampling (about half kilogram for each sample; thickness of the sampled layers less than 10 cm). Later, their grain size

distribution was determined by the mechanical method (sieving). In Figure A4, the $\varphi$ scale was used to characterize the distribution [76–78].

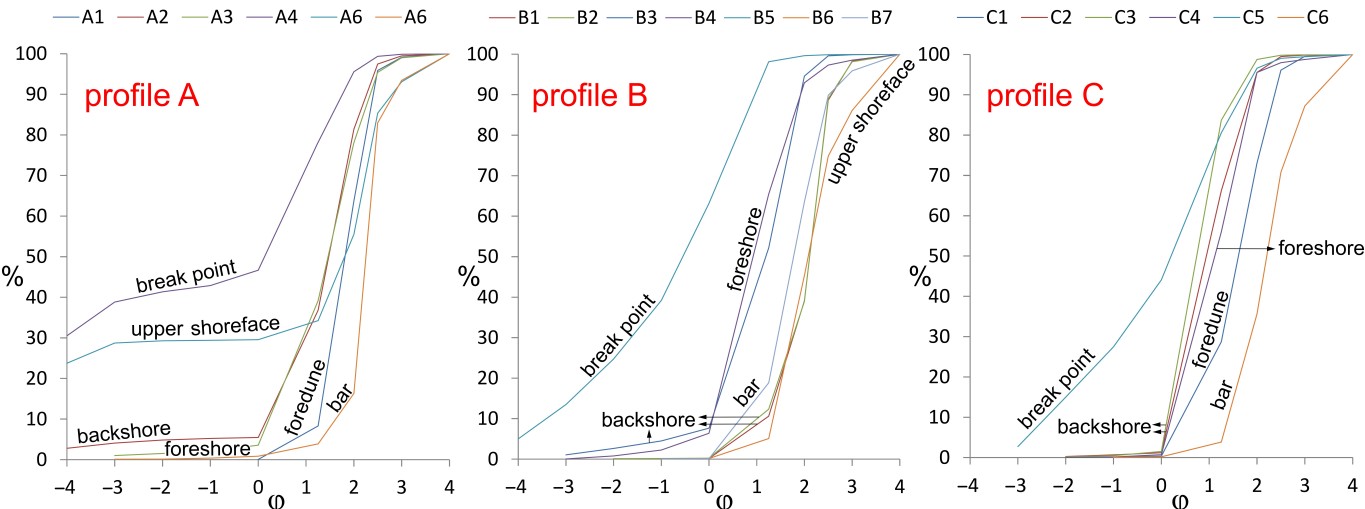

**Figure A4.** Grain-size distribution of the samples gathered at the profiles A–C, cf. Figures 7 and 8; (−4 to −3 $\varphi$, 16–8 mm, medium gravel; −3 to −2 $\varphi$, 8–4 mm, fine gravel; −2 to −1 $\varphi$, 4–2 mm, very fine gravel; −1 to 0 $\varphi$, 2–1 mm, very coarse sand; 0 to 1 $\varphi$, 1–0.5 mm, coarse sand; 1 to 2 $\varphi$, 0.5–0.25 mm, medium sand; 2 to 3 $\varphi$, 0.25–0.125 mm, fine sand; 3 to 4 $\varphi$, 0.125–0.0625 mm, very fine sand).

As reported above (see Section 4), the larger grain size was found at the base of the foreshore, in correspondence to the break point. Here, the very coarse sand is the dominant class, even if the break point of the profile A shows a significant amount of gravel (Figure A4). The grain size decreases both landward and seaward up to the fine sand class in correspondence to the dune toe and the submerged bar, respectively. The samples B1 and B2 are finer compared with the other samples taken on the backshore. This is probably the result of the eolian landward transport during the dry/warm season. Finally, it must be noted that some finer samples are mainly constituted by black minerals of volcanic origin.

**Appendix D. Additional Geomorphological Observations**

Some geomorphological insights are provided below in the perspective of further data collection and analysis. With regard to the enhanced beach and dune toe erosion at the NW side of the northwest sector (Section 4.4.1, Figure 12), it is a process that began at least 20 years ago and which in recent years has been accelerated by swash waves (Figure A5).

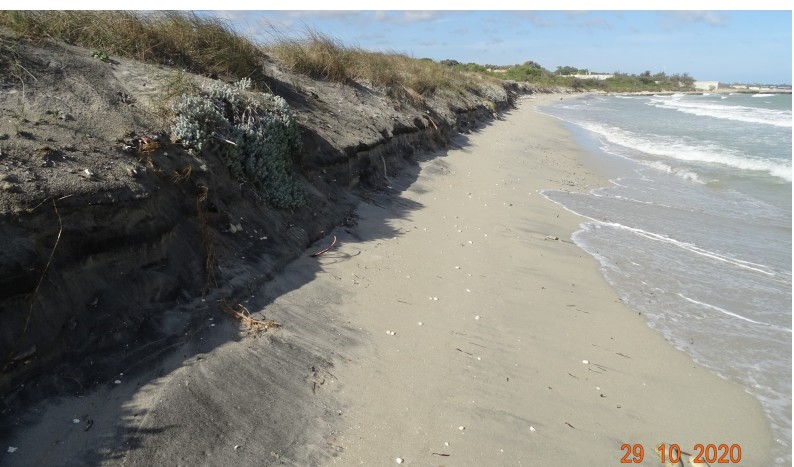

**Figure A5.** Dune toe erosion due to swash action (northwest sector).

The erosion process of the above-mentioned stretch of beach is documented by profiles measured in different time (Figure A6).

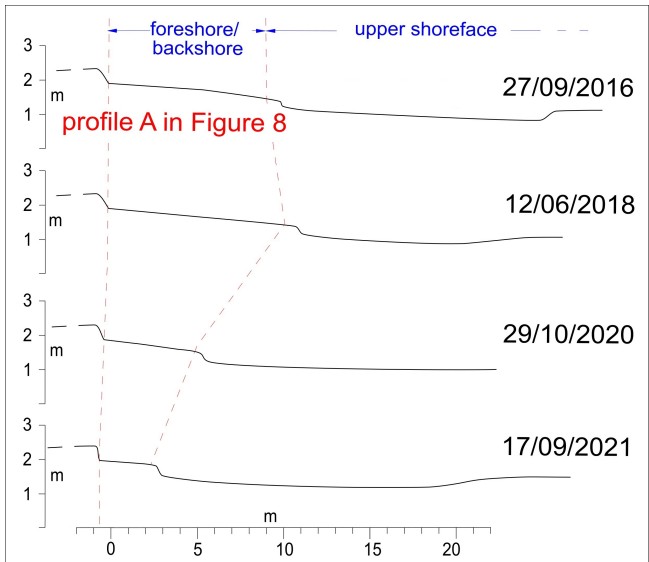

**Figure A6.** Time series of beach profiles measured along the profile A of Figure 8 (vertical scale is twice the horizontal one); dune toe apparently retreated by about 1 m. After 2021, a partial beach recovery was observed.

The foredune appears more stable in different points of the central sector, especially where the breakwater were built. Here, as a matter of fact, the dune height exceeds 2 m, and the pioneer vegetation is bushier and wider than in the NW sector (Figure A7). However, since the central sector is proved to be particularly prone to undergo dune breaking and washover accretion (Sections 4.4.1 and 4.4.2), detailed field investigations must be performed especially before and after storms.

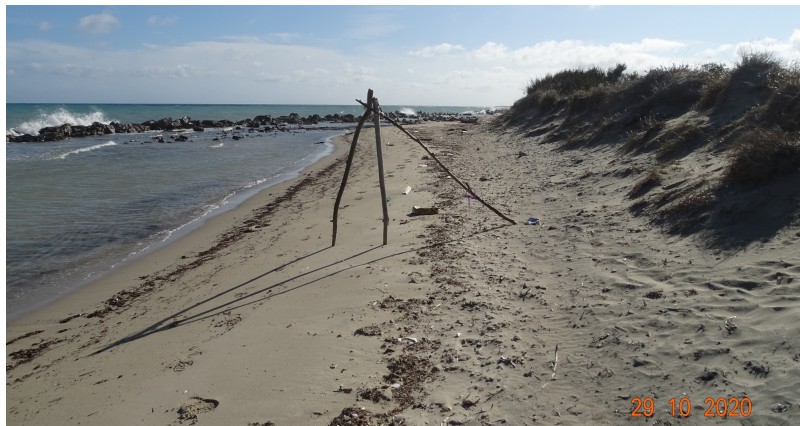

**Figure A7.** Foredune shape at a stretch of coast protected by breakwater (central sector); vertical wooden rod is about 2.5 m high.

Additionally, the southeastern sector of the lagoon barrier will need to be monitored. This stretch of coast was considered stable up to some years ago. However, washover fan accretion has been recognized by satellite image analysis (Figure 16) near Ponte di Carlo, which delimits the central and southeastern sectors of the Cesine Lagoon (see Figure 10 for location).

Since the water lagoon depth is very shallow (Section 2), the lagoon shoreline can significantly change position due to tidal oscillations in some points. This is especially evident along the edges of the washover fans where, between the high and low tide, the

land–water interface can move even tens of meters. However, the high tide shoreline is always highlighted by vegetation. Such on-site field observations are randomly documented by satellite images (Figure A8). As can be observed in Figure A8, the washover fan A (see Figure 10 for location) started to accrete between 2010 and 2012. In 2017 satellite image, its lagoon shoreline seems landward prograded in comparison with 2015. However, the 2020 and 2021 boundaries roughly resemble the 2015 one. The washover fan B slowly accreted between 2015 and 2017, accelerating the landward progradation between 2017 and 2021. Since then, its shape remained almost unchanged as for the washover fan C (Section 4). The only remarkable process observed on site during 2022 and 2023 is the deposition of a thin layers (few centimetres) of very fine black volcanic sediments due to eolian transport (see also Figures 14 and 15).

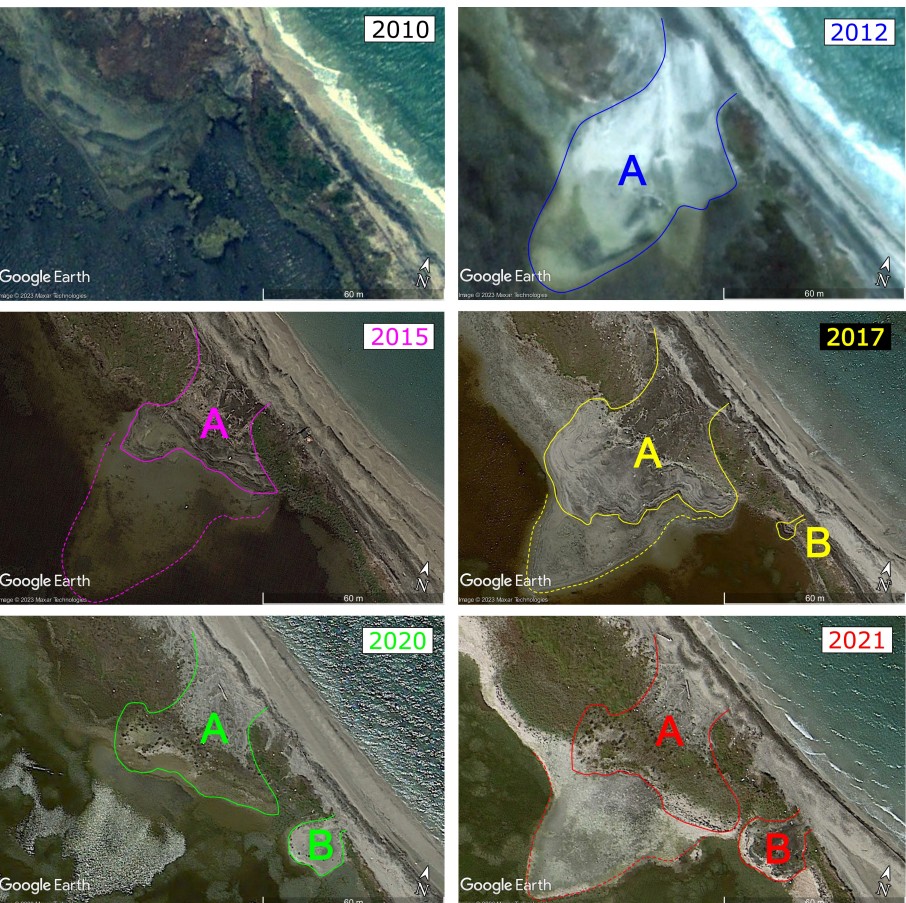

**Figure A8.** Formation and accretion of the washover fans A and B (cf. Figure 10). The dotted lines mark the submerged edge during high tide conditions. Note the black sediments that mark the shape of the backshore and the sea and lagoon shorelines (40°21′25.14″–40°21′29.31″ N, 18°20′26.17″–18°20′33.10″ E; eye elevation of 128 m).

Seasonal changes have been observed for the gravel beaches (Section 4.4.3). They form the interface with the sea during the CW season (Figure 17) but are covered by middle to very fine sand transported by waves and wind throughout the WD season, thus being sometimes indistinguishable from the sandy beaches in satellite images (Figure A9).

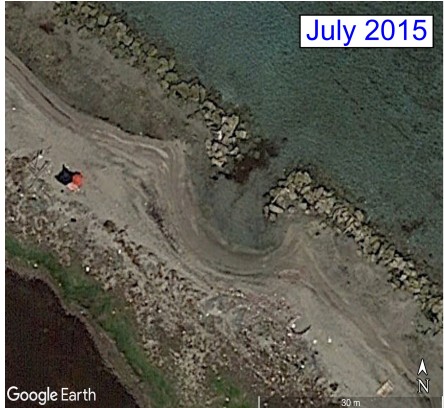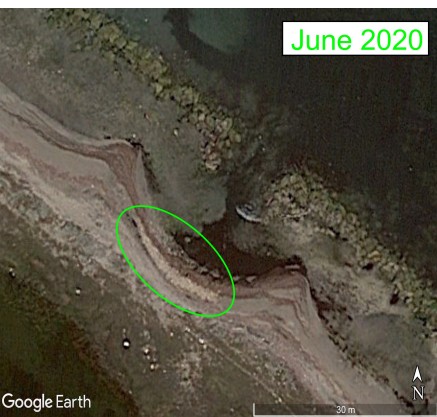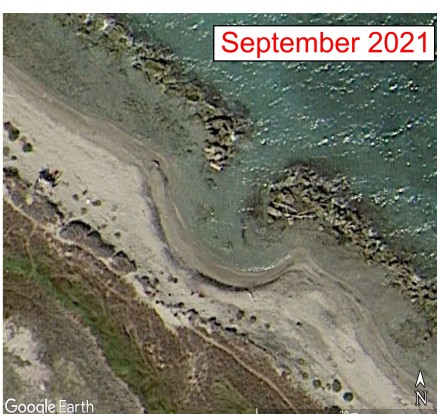

**Figure A9.** Some stages of the gravel beach of Figure 17. Such a deposit is cropped out in June 2020 (green curve), while it was covered by sand in July 2015 and September 2021 (40°22′02.28″–40°22′04.18″ N, 18°19′36.68″–18°19′38.19″ E; eye elevation of 62 m).

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
