# Peer review of "Wind–Wave Conditions and Change in Coastal Landforms at the Beach–Dune Barrier of Cesine Lagoon (South Italy)"

_climate, doi:10.3390/cli11060128_

Round 1

Reviewer 1 Report

This MS tries to study the geomorphological change of a barrier, and it has found that several coastal barriers experienced significant erosion and change in shape. As a reader, he may find that there are some problems that are difficult to understand. Thus the authors may need to improve the writing and logicality after considering the questions as follows:

1.      Based on the title of this manuscript, it seems that  wind-wave conditions are  the main factors to control the geomorphological change of the barrier. However, this MS did not show that how the change of the wind and wave influence the geomorphological change? For example, it did not indicate the relationship between the wave change and beach geomorphological change. Do they have quantitative relationship?

2.      This MS emphasize publicly available sources and simple and inexpensive methods. However, it seems that it is not very easy to access the wind and wave data for a long time. By the way, this study did not measure the beach profile during the field work?

3.      This study does not provide sufficient reasons to explain the causes of beach geomorphic changes, such as beach erosion, which is entirely caused by wind and waves. So what changes have occurred in wind and waves?

Minor editing of English language required

Author Response

General comment. This MS tries to study the geomorphological change of a barrier, and it has found that several coastal barriers experienced significant erosion and change in shape. As a reader, he may find that there are some problems that are difficult to understand. Thus the authors may need to improve the writing and logicality after considering the questions as follows.

Reply: We agree with your comments. The first version of the MS was a bit cryptic with respect to the objectives. Thus, some changes and additions have been made to improve the clarity and logicality of the MS. They include a better definition of the aims in the Introduction and short introductive comments in sections 3, 4 and 5.

Comment 1. Based on the title of this manuscript, it seems that wind-wave conditions are the main factors to control the geomorphological change of the barrier. However, this MS did not show that how the change of the wind and wave influence the geomorphological change? For example, it did not indicate the relationship between the wave change and beach geomorphological change. Do they have quantitative relationship?

Reply: It is right: "wind-wave conditions are the main factors to control the geomorphological change of the barrier". However, the original title of the MS could cause some misunderstanding. The MS intends to give a statistical description of the wind wave conditions in the 7-years period together with an analysis of the coastal processes in the same period, then trying to infer some relations where possible. Consequently, we have changed the title as follows: “Wind-wave conditions and change in coastal landforms at the beach-dune barrier of Cesine Lagoon (South Italy)”.  The writing and logicality of the MS has been first improved by editing the Introduction (please see lines 26-31 and 34-38 of the revised version). It is mentioned now that the MS aims to: “1) define the wind-wave conditions in which geomorphological changes occur; 2) infer the direction of the net longshore transport of sediment; 3) detect the geomorphological processes that shape coastal landforms, suggesting relations with wind and wave parameters”. Due to the misleading first title of the MS, we gave the impression of being able to establish the relationships between the change in wind-wave characteristics (not reasonable, however, in an only 7-years study) and the geomorphological one, as you commented. Actually, this is not yet possible since, as is then exposed in the MS, the meteorological base in the Cesine area has been in operation since 2021 (please note that we have used the Brindisi data set as a proxy for the study coast, see 4.1.1. Wind Statistics). Moreover, the aforementioned station is not located on the shoreline but rather inland, and its measurements are partially affected by the presence of a pine forest (Appendix B). At the moment we obtained only qualitative results. Due to this situation, it is our intention to make measurements on the shoreline with a weather station in the next data collection. We hope in this way to obtain quantitative relationships.

Comment 2. This MS emphasize publicly available sources and simple and inexpensive methods. However, it seems that it is not very easy to access the wind and wave data for a long time. By the way, this study did not measure the beach profile during the field work?

Reply: No wind data are available at the Cesine location before 2021, and data for this only available year are partially unreliable, as mentioned above and in the MS. A dedicated measurement campaign is being organized for the next years. Despite of the lack of data sets covering 2015-2021 at the case study, the used proxy (the Brindisi data sets) allows to define the main features of the wind-wave conditions in a realistic way. As regard the beach profiling, several measurements were made along different transects using spirit leveling. In the MS, they have been then used as a control in the satellite images interpretation. However, given your comment, in the revised version we have inserted a new figure (the Figure A6 in Appendix D) as an example of such a field work.

Comment 3. This study does not provide sufficient reasons to explain the causes of beach geomorphic changes, such as beach erosion, which is entirely caused by wind and waves. So what changes have occurred in wind and waves?

Reply: As reported in literature on the study area, the contribution of other natural and anthropic factors on geomorphological changes is substantially negligible in the considered short term time. If on one hand tectonic subsidence and changes in sea level and sediment supply can produce significant effects only on medium-long terms, on the other hand, no new human-induced process started after the building of the breakwaters in the early 2000s (lines 422-427 of the new version of MS). During the period of observation, the barrier system shaping was mainly shown by dune ridges, washover fans and gravel beaches, as we have explained in the MS. Erosion processes (but also the depositional ones) at the Cesine Lagoon barrier system is a target that has been tried to understand in its main characteristics so far (please, see lines 364-368,  379-383, 490-491, and 588-592), but that we would better explore in the future by further data collection and analysis. For the sandy beaches of the study coast, two already known processes (see cited works) agree with our results: the annual progradation/retrogradation cycle (lines 92-93, 185-189, and 540-542); and the long-term erosional trend affecting the dune-beach barrier as well as the whole major physiographic unit (lines 490-493 and cited literature). However, the incidence of cross-shore processes during storms has not yet been determined. We hope to find answers soon. The main results achieved so far are now better outlined in section 5. They are: a) the effects of coastal flooding can be connected to the possibility of a 1-year return time of storm surge of about 1 meter height (lines 442-444); b) the overall apparent southwards sediment transport is in agreement with the prevailing northerly winds (lines 445-447); c) the done toe retreat due to wave run-up can be associated to several meters height significant waves with return time of about 1-year (lines 459-461). We believe that from these findings it is possible to proceed with vulnerability analysis and comparative studies with other lagoon barriers with a climatological perspective.

Your comments were very valuable. Thank you for your time and effort to comment on our Ms.

Reviewer 2 Report

The satellite images are taken from Google Earth It is not clear which satellite with high resolution the authors are supposed to use. even all images without coordinates.

Most of the protection measures show accretion on the updrift current side and erosion at the downdrift side the case similarly found in this study for the gravels.

I recommend looking at the littoral drift current direction as well the change in hydrodynamic resulting due to the detached breakwater construction sometimes producing unexpected patterns of sedimentation especially if you have an eddy current at the updrift

Author Response

Thank you for your time and effort to comment on our work. We have first improved the research design and the result description (please see lines 134-140, 182-210, 217-226, 358-368, and 396-399 of the revised version).

Comment 1. The satellite images are taken from Google Earth It is not clear which satellite with high resolution the authors are supposed to use. even all images without coordinates.

Reply: The used high resolution satellite images are the ones taken by Landsat 8 as is now mentioned at the line 184 of the revised version of the manuscript. The coordinates of the images are in the captions of each figure.

Comment 2. Most of the protection measures show accretion on the updrift current side and erosion at the downdrift side the case similarly found in this study for the gravels.

Reply: This subject was not in our original research design. However, due to the extensive presence of defense structures in the study coast, we have added some considerations in the revised version of the manuscript (please, see lines 364-368).

Comment 3. I recommend looking at the littoral drift current direction as well the change in hydrodynamic resulting due to the detached breakwater construction sometimes producing unexpected patterns of sedimentation especially if you have an eddy current at the updrift.

Reply: In the absence of direct wave measurements, a quantitative analysis of the littoral drift as well as the whole coastal hydrodynamics would require a connection between the local wind statistics and the wave regimes, that is not trivial, together with a definition of the breaker lines and analysis of the near shore seabed. This program requires expensive observation studies to be made, we hope, in the next years. As you mentioned, this program should consider the effects of the detached breakwaters and possible localized current disturbances should be considered.

Round 2

Reviewer 1 Report

NO